# Evaluating turnaround time to improve clients' satisfaction in the tuberculosis reference laboratory in Douala

Teyim Pride Mbuh [1,2]*, Anya Priscilla Amveilla[1,3], Patricia Mendjime[1,3], Valérie Flore Donkeng Donfack[4], Mohamed Youssouf Mfouapon[1], Hamada Beloko[2], Bisso Ngono Annie Prudence[5], Esso Linda[1,3], Georges Alain Etoundi Mballa[1,3]

1 Field Epidemiology Training Program, Yaoundé, Cameroon, 2 Tuberculosis Reference Laboratory Douala, Cameroon, 3 Directorate for the Fight against Disease, Epidemics and Pandemics, Ministry of Public Health, Yaoundé, Cameroon, 4 Centre Pasteur Cameroun, Yaoundé, Cameroon, 5 Programme National de Lutte contre la tuberculose Cameroun, Yaoundé, Cameroon

* teyimpride@yahoo.com

## Abstract

### Introduction

Delivery delays of laboratory results can result in death and/or economic loss to both the patient and the health system. Data is limited regarding turnaround time for tuberculosis testing in Cameroon. We evaluated intra- turnaround time in tuberculosis diagnosis, identified root causes, and brainstormed solutions to improve patient satisfaction.

### Materials and methods

In this cross-sectional descriptive study, turnaround time for the pre-analytic phase was set at 3 hours, the analytic phase at 11 hours, and the post-analytic phase at 10 hours. The overall turnaround time was 24 hours. We used the Fishbone method of problem analysis involving the personnel of the Tuberculosis Reference Laboratory-Douala, to identify root causes. We brainstormed using the "Whys" causes of turnaround time failure during the process of tuberculosis diagnosis by GeneXpert MTB/RIF

### Results

We analyzed samples from 526 clients. The median turnaround time was 45 (Range: 2-120) hours. A total of 216(41.1%) clients had a turnaround time failure. The turnaround time failures in the pre-analytic, analytic, and post-analytics phases were 64(29.6%), 128(59.3%), and 12(11.1%) respectively. Overall, 19 root causes of turnaround time failure were identified and grouped into six categories, namely: equipment, administration, technical-staff, environment, material, and method. Equipment

**Data availability statement:** The datasets used and/or analyzed during the current study are available with the manuscript, since it is uploaded as supplementary information.

**Funding:** The author(s) received no specific funding for this work.

**Competing interests:** The authors have declared that no competing interests exist.

maintenance (defective or non-calibrated modules) was the critical cause of turnaround time failures accounting for 86 (39.8%) of the overall turnaround time failure.

## Conclusion

turnaround time in tuberculosis diagnosis is longer than expected, retarding patient care management. Focusing on equipment maintenance enhances the intra-laboratory testing process, thus improving overall patient satisfaction. The need for further studies to incorporate the extra-laboratory turnaround time in assessing the overall turnaround time is imperative.

## Introduction

In 1999, the Institute of Medicine (IOM) in their publication" to Err Is Human" [1], defined medical errors as the inability to carry out specific actions the ought to have been performed as planned or when planned. They qualified most of the medical errors to defective quality management systems which could be avoided by conceiving and monitoring safer healthcare systems and processes that diminish and avert errors, and encourage better outcomes [1]. Though numerous medical errors can be avoided, many healthcare systems, do not implement best practices and provide optimal-quality medical care, this is particular in resource-limited settings. Most medical decisions are made on the basis of laboratory findings. So, clinical laboratory findings must be accurate and timely [2]. The time interval between the specimen reception in the laboratory and the result delivery is known as laboratory waiting time or turnaround time(TAT) [3,4].

The long time patient and clinicians spends waiting for their results is often disappointing. A strong and inverse relationship has been demonstrated between patient satisfaction and result delivery TAT in an ambulatory setting [5]. Early results delivery can lead to prompt treatment which will shortened hospital stay and become an important factor that can help patients to reduction their expenses [6]. Clinical laboratories, most often, focus on the accuracy and reliability of the test results but pay very little attention to how prompt these results are released [7].

TAT is a measure of timeliness and is usually considered as an indicator for laboratory efficiency [8]. TAT is crucial form both a medical and commercial point of view. This has been explained by the reaction of most people to life situations. Most of them are usually impatient requiring that things be done timely including their clinical laboratory results. This may explain why some people prefer health facilities with the reputation of providing proper diagnosis, treatment, and management of their health problems without necessarily compromising the notion of timely services. Late result delivery can be fatal to patients [9]. Delayed TAT may be caused by factors attributed within the laboratory or out of the laboratory. TAT associated to factors within the laboratory itself are usually termed Intra-laboratory TAT while those associated to factors out of the laboratory are termed extra-laboratory TAT

The laboratory pre-analytical TAT phase refers to the time interval between the requisition of a test and the sample reaching the hands of laboratory staff. The

analytical phase is that between the beginning of the analysis and the definitive test results. The post-analytical period is the time between result printing, verification and the physician actually receives the results. Among these three phases, other studies have shown that the pre-analytical and post-analytical phases accounts to almost 96% of the TAT [3]. These findings may vary from one health facility to the other depending on the infrastructure of the institution, degree of automation in the health facility, the experience and the skills of the staffs [10,11].

Current, data is limited on the TAT for tuberculosis (TB) testing in Cameroon. This study aimed at evaluating the intra-TAT in the diagnosis of TB in the tuberculosis reference laboratory Douala (TBRL-D'la). The outcomes of this study will be useful to improve patient satisfaction by highlighting possible causes of delay in laboratory results delivery.

## Method

We conducted this cross-sectional descriptive study in the TBRL-D'la-Cameroon on all suspected TB specimens that met the TBRL-D'la sample acceptance criteria. At the reception, all samples were given a unique laboratory identification number. This identification number served as the sample's identity throughout the testing process. Patient identification was therefore unknown by the research team throughout the intervention period. We collected data to evaluate turnaround time on a daily basis as the sample moved through each stage of analysis. At each stage, the time the testing activity began and when it ended was documented. Only samples received by 8:00 am to 2:00 pm from the 1st March to 30th April 2022 for TB diagnosis by GeneXpert MTB/RIF Ultra were considered. The laboratory activities were divided into three phases (pre-analytic, analytic, and post-analytic). The pre-analytic phase was limited to specimen reception and sorting; the analytic phase was limited to specimen testing; and the post-analytic phase included result reporting, biological validation, entering patient information into the electronic database, and printing of results.

Brainstorming among the TBRL-D'la team (with academic certificates of Ph.D, Masters, and First degree) was the tool used to identify the root causes for TAT delay. The quality manager of the TBRL-D'la acted as the facilitator. The 5-Whys problem investigation method was used to drill down to the root causes. The facilitator distributed stickers to all members of the team and asked the first why question: "What are the factors responsible for a TAT of more than 24 hours in our laboratory?" Everyone on the team wrote down an idea on the sticker. Each member of the team was given a minute to share their idea. Each idea was debated. The facilitator proceeded by asking "why" four additional times, restructuring each "why" question in response to the answer the team had just provide to the preceding question. The root cause of the delay was identified when asking "why" resulted to no additional useful responses, and the team couldn't proceed. The team them acted on this point in attempt to resolve the identified root cause.

The identified root causes were represented on a fish-born diagram. The time interval spent by each specimen at each testing phase was measured in hours. The acceptable time for the pre-analytic phase was 3 hours, for the analytic, it was 11 hours, and for the post-analytic phase, it was 10 hours. The overall TAT was set at 24 hours. TAT was expressed as proportions. This was a noninvasive study. We didn't have direct contact with the patients, their biological information or nor specimens. This was a noninvasive study. We didn't have direct contact with the patients, their biological information or their specimens. We simply observe the time the Tuberculosis Reference laboratory staffs used in treating patient specimens so we didn't obtain participants' consent. We also did not judge it mandatory to obtain an ethical clearance from an Institutional Review Board (IRB) for this same purpose. We however obtained administrative clearance from the Ministry of Public Health with decision number **N° 2346/L/MINSANTE/SG/CCOUSP** to carry out this study.

The data obtained from this study was entered, analyzed, and expressed as frequencies and percentages in Microsoft Excel 2016.

## Results

Overall, 526 clients' specimens were received for TB testing by GeneXpert in the months of March and April. The median TAT of samples in this study was 45 (1–120) hours. A total of 116 and 100 of our clients in March and April, for a total of

216 (41.1%), waited for more than 24 hours before their results were ready. The pre-analytic, analytic, and post-analytic phases of TB testing accounted for 64(29.6%), 128(59.3%) and 24(11.1%) TAT failure respectively. Overall, 19 root causes of TAT failure were identified. Most, nine (47.4%) of these causes were completely out of the laboratory personnel's control, six (3.6%) were partially under their control, and four (21.1%) were completely under their control as illustrated in Fig 1.

Equipment and material-related causes of TAT failure accounted for 150 (69.4%) of all causes of TAT failure. The most predominant cause of TAT failure was equipment-related (39.8%). All TAT failures attributed to the analytic phase were either associated with defective modules 80 (93.0%) of the GeneXpert equipment or these modules not being calibrated 6 (7%). Material-related causes 64 (29.6%) were the second leading cause of failing TAT. Material-related causes were attributed to poorly filled or incomplete request forms 50 (78.1%) and poor sample quality 17(21.9%). The least associated causes of TAT failure were attributed to administrative reasons six (2.8%), as illustrated in Table 1.

## Discussion

In today's modern world, where laboratory diagnosis is shifting toward point-of-care testing, the laboratory needs to deliver prompt and accurate test results to guarantee high-quality diagnostic services. TAT for laboratory reports is now a yardstick indicator of the quality of services it renders, and many medical doctor use it to evaluate the level of a laboratory's efficiency and effectiveness [12]. The rate at which laboratory results are issued out to both the patients and the clinicians impacts both the patient outcome and the overall performance of that laboratory. We analyzed client samples received in the laboratory during the study period to generate relevant information about the TAT, a marker for their satisfaction.

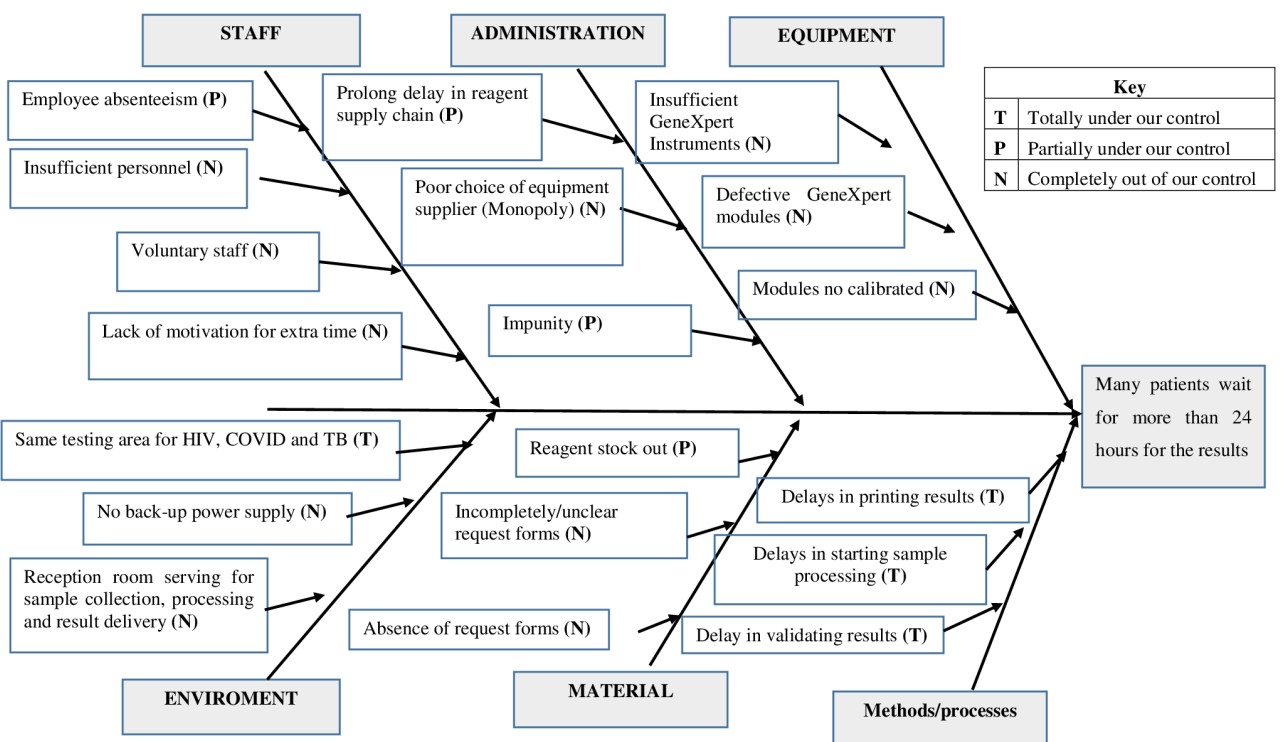

**Fig 1. Root causes for turnaround time failure in the tuberculosis reference laboratory Douala in March-April, 2022.**

**Table 1. Frequency distribution of causes of turnaround time failure in the tuberculosis reference laboratory in March to April 2022.**

| Causes of turnaround time failure | Frequency | Percentage |
|---|---|---|
| Administration | 6 | 2.8 |
| Environment | 12 | 5.6 |
| Method | 20 | 9.3 |
| Staff | 28 | 13.0 |
| Material | 64 | 29.6 |
| Equipment | 86 | 39.8 |
| Total | 216 | 100 |

In this study, 41.1% of the results of samples received in the TBRL-Douala for TB diagnosis by GeneXpert MTB/RIF Ultra were ready after the expected 24-hour waiting period. Prolonged result delivery time may delay clinical diagnosis and impact the patient's management. This has been proven to prolonged the time patient stay in the hospital and consequently increased their financial costs [8]. The consequences of prolonged TAT have been found to be more severe with respiratory tract infectious diseases and other emergency service setting. The pre-analytic, analytic, and post-analytic phases of TB testing accounted for 29.6%, 59.3%, and 11.1% of TAT failures, respectively. These findings were similar to those obtained in New Delhi, India [13] where the pre-analytic, analytic, and post-analytics phases to complete different steps of sample processing in an auto analyzer were 30.6%, 42.6%, and 26.9%, respectively. Other studies [14–16], however, found that 50–75% of the laboratory TAT failures were attributed to the pre-analytic phase because this phase is not in the direct control of laboratory personnel and also involves multiple steps and laboratory personnel. In our study, 90.7% of the delay during the analytic phase was attributed to equipment-related delays, as the laboratory was functioning on five out of 12 GeneXpert modules lodged in three instruments. Other studies have demonstrated that the equipment breakage are the most recurrent cause of laboratory result delays [12], reagent stock out has been the second on the list [17]. Other causes of a TAT failures have been attributed to the inability to modify work schedules to organize available manpower and an insufficient manpower at work [17]. Moreover, the overlap of activities caused by supportive supervision, intensive training and mentorship, of students on internships kept laboratory personnel very busy and could have delayed our TAT. Administrative tolerance was also a major cause for TAT failure in this study, given that in 2020, the maintenance of all (approximately 75) the GeneXpert instruments in the entire country was given to a subcontracting company that happens to have a limited number of trained personnel. These few people have to travel over the national territory to maintain all the GeneXpert instruments, even for minor problems. To make things worse, this company doesn't have a single buffer stock of spare parts or replacement modules in the country. Each time a fault was signaled, a long time was wasted waiting for the local company to come onsite and certify the fault. An even greater amount of time was spent waiting for the spare part to travel from the manufacturing country.

## Limitations

This study was limited to the evaluation of intra-laboratory turnaround time. We therefore recommend that a similar study be extended to cover both the intra- and extra-laboratory turnaround times to better understand the causes of prolonged patient waiting times to acquire laboratory service for TB diagnosis in the littoral region of Cameroon.

## Conclusion

Monitoring TAT for TB diagnosis is crucial as a marker of patient satisfaction and identification of possible causes for delay in laboratory results delivery. This study revealed that GeneXpert maintenance, especially delays in module replacement, was responsible for most of the analytic testing phase TAT failures. We encourage open policies with competition for all laboratory equipment and the availability of a buffer stock of essential spares parts and replacement modules in-Country.

## Supporting information

**S1 File. TAT data for plos one.**
(XLSX)

## Acknowledgments

The authors thank all the Tuberculosis Reference Laboratory Douala staffs who provided us with very useful information which enabled us identify root causes for TAT failure.

## Author contributions

**Conceptualization:** Teyim Pride Mbuh, Anya Priscilla Amveilla, Patricia Mendjime, Mohamed Youssouf Mfouapon, Bisso Ngono Annie Prudence, Esso Linda, Georges Alain Etoundi Mballa.

**Data curation:** Patricia Mendjime, Valérie Flore Donkeng Donfack, Hamada Beloko.

**Formal analysis:** Teyim Pride Mbuh, Patricia Mendjime, Valérie Flore Donkeng Donfack, Mohamed Youssouf Mfouapon, Hamada Beloko.

**Investigation:** Mohamed Youssouf Mfouapon, Hamada Beloko.

**Methodology:** Teyim Pride Mbuh, Anya Priscilla Amveilla, Patricia Mendjime, Valérie Flore Donkeng Donfack, Mohamed Youssouf Mfouapon, Hamada Beloko, Bisso Ngono Annie Prudence, Esso Linda, Georges Alain Etoundi Mballa.

**Supervision:** Anya Priscilla Amveilla, Patricia Mendjime, Bisso Ngono Annie Prudence, Esso Linda, Georges Alain Etoundi Mballa.

**Validation:** Teyim Pride Mbuh, Valérie Flore Donkeng Donfack, Hamada Beloko, Bisso Ngono Annie Prudence, Esso Linda.

**Visualization:** Hamada Beloko.

**Writing – original draft:** Anya Priscilla Amveilla, Patricia Mendjime, Valérie Flore Donkeng Donfack, Mohamed Youssouf Mfouapon, Hamada Beloko, Georges Alain Etoundi Mballa.

**Writing – review & editing:** Teyim Pride Mbuh, Mohamed Youssouf Mfouapon, Hamada Beloko, Bisso Ngono Annie Prudence, Esso Linda.

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
