## [Editor Report · Decision Letter 0]

16 Jan 2025

PONE-D-24-50358Evaluating Turnaround Time to Improve Clients' Satisfaction in the Tuberculosis Reference Laboratory in DoualaPLOS ONE

Dear Dr. Mbuh,

Thank you for submitting your manuscript to PLOS ONE. After careful consideration, we feel that it has merit but does not fully meet PLOS ONE’s publication criteria as it currently stands. Therefore, we invite you to submit a revised version of the manuscript that addresses the points raised during the review process.

We look forward to receiving your revised manuscript.

Kind regards,

Talkmore Maruta, PhD

Academic Editor

PLOS ONE

Journal Requirements:

3. In the online submission form, you indicated that the datasets used and/or analyzed during the current study are available from the corresponding author teyimpride@yahoo.com upon reasonable request. 

4. Please remove all personal information, ensure that the data shared are in accordance with participant consent, and re-upload a fully anonymized data set. 
---

## [Author Response · Author response to Decision Letter 1]

12 Feb 2025

A rebuttal letter for the manuscript titled: Evaluating Turnaround Time to Improve Clients' Satisfaction in the Tuberculosis Reference Laboratory in Douala submitted to PLOS ONE reference PONE-D-24-50358

SN Reviewers’ preoccupation Response provided

1 Please ensure that your manuscript meets PLOS ONE's style requirements, including those for file naming.

Thanks, the manuscript has been reformatted to meets PLOS ONE's style requirements, including those for file naming.

2 Please provide additional details regarding participant consent. In the ethics statement in the Methods and online submission information, please ensure that you have specified what type you obtained (for instance, written or verbal, and if verbal, how it was documented and witnessed). If your study included minors, state whether you obtained consent from parents or guardians. If the need for consent was waived by the ethics committee, please include this information.

This was a noninvasive study. We didn’t have direct contact with the patients, their biological information or their specimens. We simply observed the time laboratory staffs used in treating patient specimens so we didn't obtain participants' consent.

This is clearly stated in the manuscript in the method (lines 119-121) and the declaration (lines 203-205)

3 In the online submission form, you indicated that the datasets used and/or analyzed during the current study are available from the corresponding author teyimpride@yahoo.com upon reasonable request.

the datasets used and/or analyzed during the current study are available with the manuscript, since it was uploaded as supplementary information. This in found in the manuscript in lines 239-240

4 Please remove all personal information, ensure that the data shared are in accordance with participant consent, and re-upload a fully anonymized data set.

The dataset shared doesn’t contain any personal information that may compromise or identify the study participant

5 Please include captions for your Supporting Information files at the end of your manuscript, and update any in-text citations to match accordingly. Please see our Supporting Information guidelines for more information: http://journals.plos.org/plosone/s/supporting-information.

A caption for the datasets used and/or analyzed during the current study was included.

6 Please review your reference list to ensure that it is complete and correct. If you have cited papers that have been retracted, please include the rationale for doing so in the manuscript text, or remove these references and replace them with relevant current references. Any changes to the reference list should be mentioned in the rebuttal letter that accompanies your revised manuscript. If you need to cite a retracted article, indicate the article’s retracted status in the References list and also include a citation and full reference for the retraction notice.

Thanks, this has been done

---

## [Decision Letter · Decision Letter 1]

17 Apr 2025

Evaluating turnaround time to Improve clients' satisfaction in the tuberculosis reference laboratory in Douala

PONE-D-24-50358R1

Dear Dr. Mbuh,

We’re pleased to inform you that your manuscript has been judged scientifically suitable for publication and will be formally accepted for publication once it meets all outstanding technical requirements.

Kind regards,

Enoch Aninagyei, PhD

Academic Editor

PLOS ONE

Additional Editor Comments (optional):

Reviewers' comments:

Reviewer's Responses to Questions

**Comments to the Author**

1. If the authors have adequately addressed your comments raised in a previous round of review and you feel that this manuscript is now acceptable for publication, you may indicate that here to bypass the “Comments to the Author” section, enter your conflict of interest statement in the “Confidential to Editor” section, and submit your "Accept" recommendation.

Reviewer #1: All comments have been addressed

2. Is the manuscript technically sound, and do the data support the conclusions?

Reviewer #1: Yes

3. Has the statistical analysis been performed appropriately and rigorously? 

Reviewer #1: Yes

4. Have the authors made all data underlying the findings in their manuscript fully available?

Reviewer #1: Yes

5. Is the manuscript presented in an intelligible fashion and written in standard English?

Reviewer #1: Yes

6. Review Comments to the Author

Reviewer #1: The authors submitted an important study topic and rationally put causes of delay in TAT. Although this is a qualitative study needs minimal analysis but expressed well caused behind prolonged. Authors responded well to reviewers comments.

7. PLOS authors have the option to publish the peer review history of their article (what does this mean? ). If published, this will include your full peer review and any attached files.

**Do you want your identity to be public for this peer review?** For information about this choice, including consent withdrawal, please see our Privacy Policy .

Reviewer #1: **Yes: ** LAYTH AL-SALIHI

---

## [Editor Report · Acceptance letter]

PONE-D-24-50358R1

PLOS ONE

Dear Dr. Mbuh,

I'm pleased to inform you that your manuscript has been deemed suitable for publication in PLOS ONE. Congratulations! Your manuscript is now being handed over to our production team.

Kind regards,

on behalf of

Dr Enoch Aninagyei

Academic Editor

PLOS ONE